# ONE-WAY PROTOTYPICAL NETWORKS

## ABSTRACT

Few-shot models have become a popular topic of research in the past years. They offer the possibility to determine class belongings for unseen examples using just a handful of examples for each class. Such models are trained on a wide range of classes and their respective examples, learning a decision metric in the process. Types of few-shot models include matching networks and prototypical networks. We show a new way of training prototypical few-shot models for just a single class. These models have the ability to predict the likelihood of an unseen query belonging to a group of examples without any given counterexamples. The difficulty here lies in the fact that no relative distance to other classes can be calculated via softmax. We solve this problem by introducing a "null class" centered around zero, and enforcing centering with batch normalization. Trained on the commonly used Omniglot data set, we obtain a classification accuracy of .98 on the matched test set, and of .8 on unmatched MNIST data. On the more complex MiniImageNet data set, test accuracy is .8. In addition, we propose a novel Gaussian layer for distance calculation in a prototypical network, which takes the support examples' distribution rather than just their centroid into account. This extension shows promising results when a higher number of support examples is available.

## 1 MOTIVATION

One- or few-shot learning is an active area of research that focuses on modeling semantic concepts with very little training data. Such models learn information about the general task of comparing queries to one or several support examples of the requested classes, which can then be applied to unseen classes. A large amount of data is still necessary for training, but this data does not need to belong exactly to the domains that may become interesting in the future, but rather to a variety of similar problems. This notion is inspired by human learning; humans can usually generalize to a new concept from just one or very few examples, using their prior knowledge of the world and related concepts.

A special case occurs when we do not need to distinguish between multiple concepts, but rather want to enable models to recognize a single new concept. In many practical scenarios, we will often be interested in just one novel class, rather than multiple at once. Application examples include detecting news or social media texts about a certain events; detecting one type of unseen object (e.g. a face) in a dataset; verification of unseen fingerprints; or outlier detection within an unseen class. In existing few-shot methods, this is commonly translated into a two-class problem with the classes "belonging to the concept" and "not belonging to the concept". As these two classes have very different types of feature distributions, this is not the most elegant solution and poses problems, especially with regards to training data selection. It also requires effort for modeling a class (the negative one) that is not of interest.

In this paper, we present an addition to few-shot learning approaches that enables generalization to a single class in order to determine whether queries belong to this (previously unseen) concept or not. This is implemented by introducing a "null" (or "garbage") class which does not need to trained to the prototypical network approach. Application of this novel model type is demonstrated on the commonly used *Omniglot* and *MiniImageNet* data sets.

## 2 RELATED WORK

### 2.1 ONE-CLASS CLASSIFICATION

The problem of one-class classification has been a topic of research for decades. It has also been interpreted as outlier detection, anomaly detection, novelty detection, and concept learning. Comprehensive overviews of "traditional" methods for solving this task are given in Khan & Madden (2014) and Chandola et al. (2009). Chandola et al. (2009) identify several different fundamental techniques, e.g. statistics, nearest-neighbour methods, clustering, and classification. Strategies can be derived from research areas such as data mining, information theory, and machine learning.

Within the machine learning domain, a common solution relies on One-class Support Vector Machines (OSVMs) [Schölkopf et al. (1999); Tax & Duin (1999)], although earlier-stage neural networks were also utilized [Hawkins et al. (2002)]. In recent years, various deep learning approaches have been suggested as well. Auto-encoders are frequently used for dimensionality reduction (corresponding to a feature extraction) of the input data in this context. Some methods then directly calculate the reconstruction error to determine whether new queries belong to the same class or are outliers (e.g. Sakurada & Yairi (2014); Chen et al. (2015)). Others add a subsequent classification network (e.g. Morton & Griffin (2016) or Erfani et al. (2016), where the auto-encoder is replaced with a Deep Belief Network). Ruff et al. (2018) take inspiration from OSVM approaches and train CNNs jointly with a classification SVM. Lately, adversarial approaches have seen attention in this domain as well, e.g. in Schlegl et al. (2017) and Sabokrou et al. (2018), where they are used in a similar fashion as the previous auto-encoders.

Direct training on a single annotated class is not implemented as frequently as the combination of automatic feature extraction and distance (or reconstruction error) calculation. Oza & Patel (2019) demonstrates a strategy for training a CNN on positive examples and artificial negative examples generated from Gaussian distributions, while Perera & Patel (2018) shows a method based on automatic feature extraction in combination with specialized loss functions and template matching. A broad overview of deep learning techniques from the anomaly detection perspective is provided in Chalapathy & Chawla (2019).

All of these approaches rely on the availability of data from the expected class for training. What is missing is a type of model that can determine whether a query belongs to a class from a small number of examples that are provided at evaluation time without the need to re-train. Such a classification is possible with few-shot learning.

### 2.2 FEW-SHOT LEARNING

The concept of one- or few-shot learning is motivated by human learning: If humans are presented with a new class of objects, they do not require more than a few examples to be able to match new instances to this class (often, just one supporting example is sufficient). This ability is, among other factors, informed by general world experience, which teaches humans to set boundaries between object classes.

Matching networks, which implement this idea, were first introduced by Vinyals et al. (2016). Training is performed on sets of support examples for a subset of possible classes plus a query example belonging to one of the classes in so-called episodes. These episodes are generated for a wide range of class permutations. In the one-shot case, only one support example is given for each class, while there are multiple ones in few-shot models.

Such a network has two input branches: One for the support examples, and one for the query example. In both branches, the inputs are run through a sub-network which performs an embedding. The embedding networks may or may not share their weights. The query's resulting embedding is then compared against the supports' resulting embeddings using a pre-defined distance metric. This results in likelihood measures for each input class, commonly implemented with a softmax. The over-all comparison metric for the task is learned implicitly as the network learns an appropriate embedding.

In the few-shot case, multiple support examples per class are treated independently of each other throughout the network. Example-related likelihoods are then summed up to obtain class likelihoods at the output. In this way, matching networks essentially implement a weighted nearest-neigbor classification. As a further development, Snell et al. (2017) introduced prototypical networks. Instead of treating all of a class' support examples independently, they compute a so-called "prototype" of the class after the embedding network. This prototype is commonly the mean vector (or centroid) of

the class's embedded support examples.When the squared Euclidean distance is used as the metric, this implements a linear classifier rather than the nearest neighbor classifier produced by matching networks.

Finn et al. (2017) goes one step further and introduces model-agnostic meta learning (MAML): An algorithm to train models that can be easily finetuned to new tasks with very few examples. This can also be applied to other few-shot methods such, as matching networks.

When applying few-shot learning to a binary problem, support examples for two classes are required: Positive examples of the class of interest, and random other examples. This amounts to a workaround, as there are no two comparable classes. Instead, one of them merely consists of counterexamples (a sort of "garbage" class). Examples in this class will have a much wider feature distribution. This makes selecting random examples difficult, as they should still be representative of this distribution, particularly when only a small number of support examples is used. It also means that computational power and time must be dedicated to a class that is not actually of interest, and as such is not very elegant. Siamese networks [Koch et al. (2015)] can be seen as an interesting way to solve this problem, as they implement a decision about whether two unseen examples belong to the same class or different ones. However, they are not able to exploit more than one support example per class like prototypical networks can.

## 3 PROPOSED APPROACH

### 3.1 ONE-WAY PROTOTYPICAL MODELS

Motivated by the counterexample issue, we take the described approach one step further: We construct a prototypical network that only requires positive support examples, from which it generates an internal embedding of the whole class. The query embedding is then compared to this, and judged to belong to the same class or not. We call this type of network a "one-way prototypical network". The sub-network for generating the internal embeddings is implemented just as in regular prototypical networks with shared weights between the support and query branches. An illustration is shown in figure 1a.

This approach presents a challenge for the distance metric: Because Cosine or Euclidean distances are not bounded, we cannot set a threshold for class belonging. In order to do this, we need a basis for comparison. We implement this by introducing a "null class" that models the whole space of possible examples without explicitly being given those examples (as in the two-class implementation mentioned above). Queries are judged on whether they are more probable to belong to the (positive) class we are interested in rather than the null class.

The question then becomes how to construct this class. As explained above, prototypical networks model classes by calculating the centroid of the class's support examples. As the model includes batch normalization layers, we can assume that the centroid of the whole latent space will be close to the zero point in all embedding dimensions. A query's distance to the centroid of the positive class is therefore compared to its norm (i.e. its distance to the zero centroid) to determine whether it is more likely to belong to the positive class or to the null class encompassing everything. Figure 1b shows a visualization where the null centroid is denoted as $c_0$. A similar idea was presented in Ren et al. (2018), which suggests the use of unlabeled data to improve prototypes. The null (or "distractor") class here is only used to identify examples not belonging to any of the trained classes, rather than training the model to explicitly distinguish between a single given class and the null class. As in regular prototypical models, Euclidean distance is used.

One issue that arises is that in the state-of-the-art prototypical models, batch normalization is not performed at the end of the convolution blocks of the embedding network, but rather after the convolution layer and before the (ReLU) non-linearity and pooling layers. The final block is therefore not quite batch-normalized. As there is some debate about the ordering of these layers anyway [Ioffe & Szegedy (2015); Mishkin], we propose reordering the block layout to a structure of: Convolution - (Leaky) ReLU - Max Pooling - Batch Normalization. An alternative to the null class solution would be a simple thresholding of the class likelihood. However, preliminary experiments showed that this threshold is very problem- and concept-dependent, and thus does not generalize well to unseen episodes.

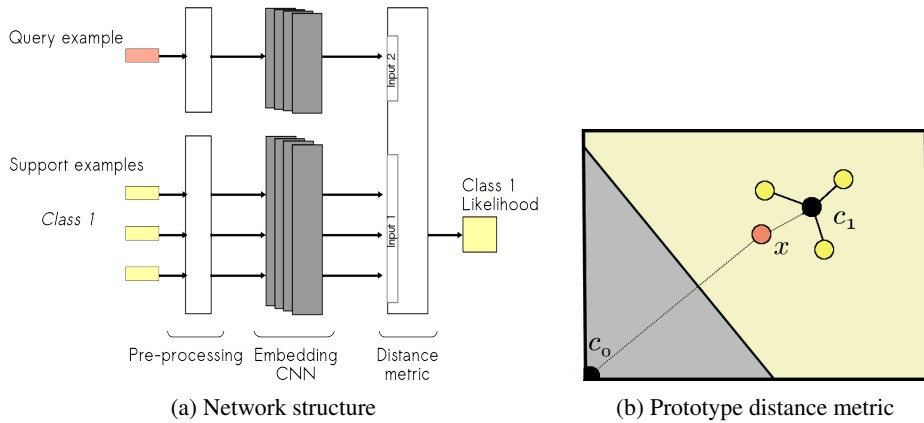

(a) Network structure        (b) Prototype distance metric

Figure 1: One-way few-shot network structure and prototype calculation

## 3.2 ONE-WAY NORMAL PROTOTYPICAL MODELS

As described, prototypical networks only take into account the centroid of a class's support examples. This is sufficient when all classes can be assumed to be roughly on the same semantic level and therefore have similar feature distribution widths. However, this may not always be true (e.g. we may encounter cases where the model attempts to distinguish between queries of the class "Bobtail cat" and queries of the class "Insect"). This problem becomes even more prevalent for the one-class case, as unseen classes may have wider or narrower distributions than unseen examples, and we do not have a basis for comparison defined by other classes.

Allen et al. (2019) suggest solving this problem by dynamically adding prototypes per class as necessary - so-called Infinite Mixture Prototypes (IMP). We propose a different extension to prototypical networks: Modeling classes by their distribution instead of their centroid only. These distributions are fitted to the class's support examples. Due to the effect of batch normalization, we assume multivariate normal distributions for this extension. A query's class assignment is then determined by its likelihood under each class's distribution. The previous embedding network is implemented just as before, except that Leaky ReLU activations become necessary instead of ReLUs to guarantee training stability. (To clarify: If the value of any of the embedding dimensions becomes exactly zero after the non-linearity, the all-over probability becomes zero as well, leading to a failure of gradient calculation during training. Leaky ReLU prevents this from happening).

This extension of prototypical networks also integrates with the one-way framework described above. Instead of assuming a centroid at zero for the null class, we now assume a multivariate normal distribution with a mean at zero and a standard deviation of 1 (this is not guaranteed exactly, though: Batch normalization enforces a centering at zero and a standard deviation of 1, but does not enforce Gaussian-ness). A visualization is provided in figure 2. (In the one-way case, this is an advantage over IMP: Preliminary experiments show that adding the one-way concept to IMP models does not lead to improvements over standard prototypical networks as the multiple positive prototypes tend to overpower the one negative prototype.)

Of course, this idea has one obvious drawback: The model requires a sufficient number of support examples to be able to determine a salient standard deviation over them, particularly as the embedding dimensions are treated independently. We hypothesize that beyond a certain number of support examples, implementing class's distributions instead of their centroids only will enable the model to distinguish between them better.

(As a side note, Fort (2017) also showed how to integrate a distribution concept into prototypical networks, calling them "Gaussian Prototypical Networks". However, in this publication, the covariance matrix is calculated for each embedded example rather than over the various support embeddings. It is then taken into account explictly in the distance metric. This allows a better handling of uncertainties, particularly for noisy data. Our focus here is somewhat different: We are more interested in obtaining additional information from the relationships between support examples).

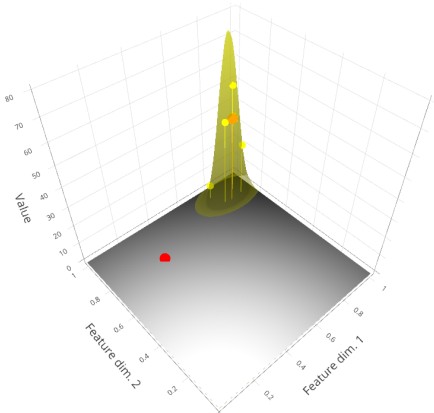

Figure 2: Normal prototype example - Support examples and estimated class distribution in yellow, null class distribution in grey; positive query in orange, negative query in red

## 4 EXPERIMENTS AND RESULTS

### 4.1 DATA AND EXPERIMENTAL DESIGN

We first test our approach on a well-known toy data set for one- or few-shot learning: The *Omniglot* image data set [Lake et al. (2015)]. This data set contains 1623 characters hand-drawn 20 times by Amazon Mechanical Turk[1] annotators. These characters come from 50 alphabets; a 30-20 split of the alphabets is suggested for training and validation. Per episode, we use 1 to 19 instances of a character as support examples, and either one of the remaining examples or a random other character as the query. As commonly done in literature, we also test the trained models on an unmatched data set: The *MNIST* handwritten digit database [LeCun et al. (1998)].

Second, we run experiments on the much more complex *MiniImageNet* data set [Vinyals et al. (2016)]. *MiniImageNet* is a subset of the larger *ImageNet* data set [Russakovsky et al. (2015)]. It contains 60,000 color images of size 84x84, split across 100 semantic classes with 600 examples each. Out of these, 80 classes are used for training and 20 for test. We use the splits provided by Ravi & Larochelle (2017). Our experiments are performed with 1 to 20 support examples per class. In both cases, pre-processing simply consists of a normalization of the input image data to a mean of 0 and a standard deviation of 1. All trainings are performed six times with random initializations and training episodes. The results are reported in terms of accuracy and averaged across runs; standard deviation between runs is generally very low ($<.01$). Each training is run for 20 (for *Omniglot*) or 10 (*MiniImageNet*) episodes, with 32,000 randomly generated episodes each, resulting effectively in 640,000 and 320,000 training episodes respectively. Adam is used as the optimizer.

### 4.2 EXPERIMENT A: SIAMESE AND TWO-WAY MODELING

In our first experiment, we test the performance of previous architectures on the one-way problem: With Siamese, matching, and prototypical networks. The former can only integrate a single support example and compare this to the query. For the latter two, we implement the one-way problem as a two-way classification, with the relevant "positive" class on the one hand, and the "negative" class on the other. Support examples are thus selected from the positive class, and randomly from the whole remaining data set. Results on the *Omniglot* and *MiniImageNet* data sets are displayed in figure 3. Supports range between 1 and 19 (*Omniglot*) or 20 (*MiniImageNet*).

The results are generally acceptable, with prototypical models always delivering higher accuracies than matching networks. This difference becomes larger the more support examples are available, as this allows the prototypical networks to model the negative class better, whereas the nearest-neighbor classification of matching networks is not strongly improved with more support examples. Accuracy of the prototypical networks ranges between .75 for one support example, and .977 for 19

---

[1] https://www.mturk.com/

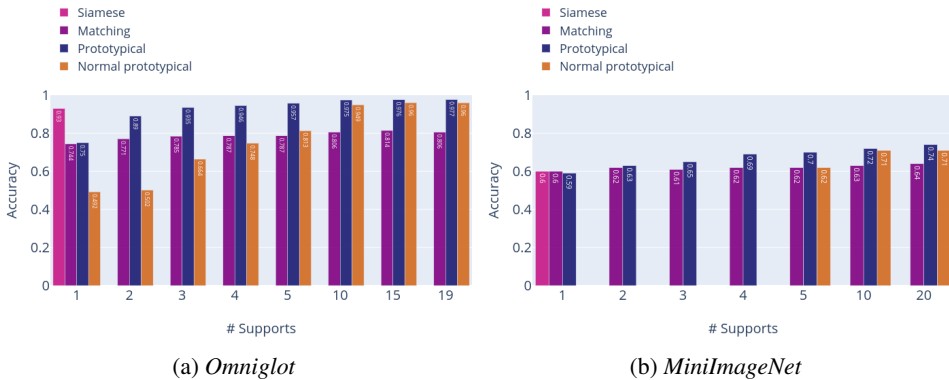

Figure 3: Accuracies of models implementing the one-way problem with two classes

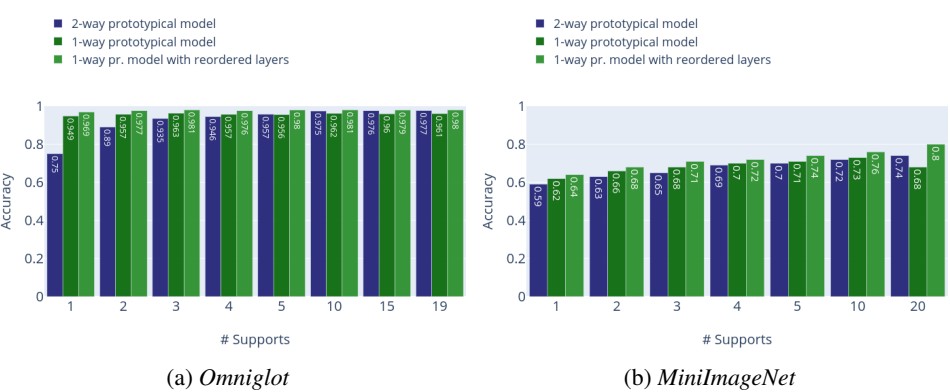

Figure 4: Accuracies of one-way prototypical models

on the *Omniglot* data set, and between .59 (one support) and .74 (20 supports) on *MiniImageNet*. Siamese networks perform well (even best on *Omniglot*) for one support example, but are quickly surpassed by prototypical networks when more are available.

The novel normal prototypical networks work well for this problem, but do not surpass standard prototypical networks. For very few support examples, results do not rise much above the .5 baseline, or training may even fail completely. This is to be expected, as the distribution cannot be determined from such a low number of supports. 5 examples appear to be necessary to make these models work at all, and 10 to make them generate competitive results.

We also test the models trained on *Omniglot* on the unmatched *MNIST* data set. Results are not shown here, but follow the same trends, just somewhat lower. The accuracy of prototypical networks is .686 for one support example, and .883 for 19.

## 4.3 EXPERIMENT B: ONE-WAY PROTOTYPICAL MODELS

We next test the novel one-way configuration for prototypical models which distinguishes between a "positive" class that we are interested in, and a "null" class that is fixed at a zero centroid as described in section 3.1. The results are shown in figure 4. The first bar in each group is the two-way result from the previous experiment for comparison. The second one displays the accuracy of the same type of model, but with the null class instead of an explicit "negative" class. As mentioned above, we also test a model with reordered layers so that batch normalization comes last before the prototype calculation; the third bar shows the results for this model.

We discover that the one-way realization works better in almost all cases. Results improve particularly in cases with few support examples. This makes sense, as few examples do not allow the two-way model to find the centroid of the negative class easily, whereas this is pre-defined in the one-way network. We also notice that the reordered model provides better results than the regular one; batch-normalizing our latent representations right before the prototype calculation therefore

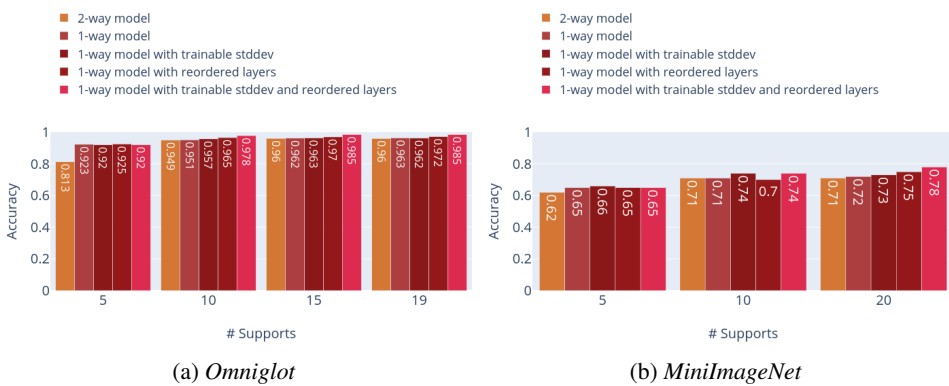

Figure 5: Accuracies of one-way normal prototypical models

appears to be important (we observe the same effect when simply adding another batch normalization layer to the standard model). With this model, accuracy on *Omniglot* rises to .969 for one support example, and to .98 for 19. On the more complex *MiniImageNet* data, accuracy becomes .64 for one support example and .8 for 20. For comparison, this re-ordering was also tested on the two-way prototypical model; no significant differences in the results were found when compared to the original layer ordering.

Once again, we also test the *Omniglot* models on *MNIST*. For 19 support examples, the accuracy is slightly worse compared to the two-class model at .796. This makes sense as the batch normalization is optimized on *Omniglot*, and a large number of supports allows for a better estimation of the negative class' distribution. However, results are better or equal for a small number of support examples (e.g. .743 for one).

## 4.4 EXPERIMENT C: NORMAL PROTOTYPICAL MODELS

Finally, we test the various configurations of our novel normal prototypical models described in section 3.2. The results are shown in figure 5. Out of these, the first bar in each group displays the two-way model accuracy from experiment A, in which the prototypes are modeled with a normal distribution. The second bar shows the results for the same model, but with a null class modeled with a unit Gaussian (mean 0, standard deviation 1) instead of the explicit negative class. Third, we allow the model to train the standard deviation of the null class instead of fixing it at 1. The fourth and fifth bars show results for similar models, but once again with the embedding layers reordered so that the last step before the prototype calculation is batch normalization.

As described in experiment A, this type of model does not produce usable results with few support vectors, and may even fail to train. For this reason, we only show results for 5 or more support vectors. With three or more for *Omniglot* or 5 or more for *MiniImageNet*, we start to notice an improvement when using the one-way implementation over the two-way version - just as in the previous experiment. Once again, reordering the embedding layers to put batch normalization last increases accuracy, but the effect is not as strong as in standard prototypical models. Allowing the model to adapt the standard deviation of the null distribution during training also helps. This is somewhat surprising because batch normalization enforces a standard deviation of 1 over all data. We suspect that this happens because the actual distribution is not perfectly Gaussian, and therefore is not modeled well by our assumptions. We tested adding a loss term to enforce Gaussian-ness, but did not see any improvemtents. This effect will require closer investigation in the future.

Over-all, the results follow the same trends as those of the standard prototypical models, but mostly remain slightly below them. However, for a larger number of support examples ($\geq 15$), they surpass them slightly in accuracy on the *Omniglot* data set. Future experiments with even more supports are necessary to determine where they become useful. Over-all, we obtain accuracies of .985 for *Omniglot* with 19 support examples, and .78 for *MiniImageNet* with 20. When testing the *Omniglot* models on *MNIST*, they yield an accuracy of .798.

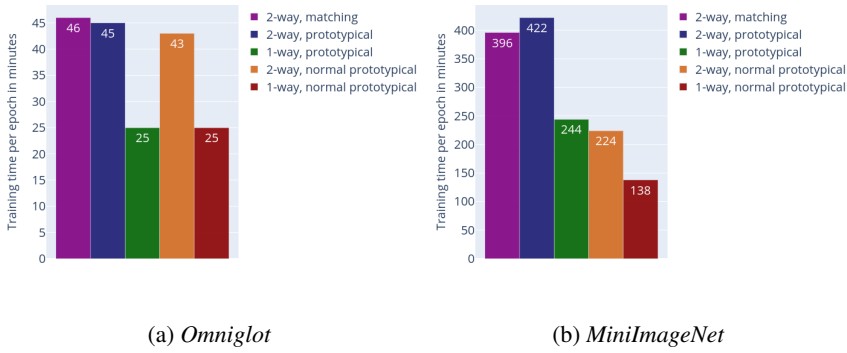

(a) *Omniglot*                                       (b) *MiniImageNet*

Figure 6: Average training time per epoch

## 4.5 EXPERIMENT D: TRAINING TIME

As an additional experiment, we analyze the training times of our various models on an NVIDIA Volta V100 with 16GB RAM. This is calculated by averaging the training times of the last three epochs of each model. Figure 6 displays the results on *Omniglot* and *MiniImageNet*. As mentioned above, each epoch consists of 32,000 episodes, and we choose the configuration with 5 support examples for comparison. The first two bars in each group correspond to the two-way matching and prototypical models from experiment A; their training times are roughly in the same range. Interestingly, the two-way normal prototypical model (fourth bar, also from experiment A) trains slightly faster on *Omniglot* and much faster for *MiniImageNet* while achieving almost the same results as the prototypical model. We are currently investigating the reasons for this.

More importantly, bars three and five show the training times for the one-way implementations. For both prototypical and normal prototypical models, training time is almost halved. This is explained by the fact that only half the support examples need to be processed, and the centroids or distributions only need to be calculated for one class instead of two in each episode. As seen above, these models also usually deliver better results than the two-way implementations, thus providing another argument for their usage.

## 5 CONCLUSION AND FUTURE WORK

In this paper, we presented approaches for few-shot learning in cases where we are only interested in a single class in each episode. Naively, this can be modeled with a two-way matching or prototypical model with support examples for the relevant, "positive" class, and random other support examples for the "negative" class. This is not very elegant as selecting negative examples is difficult, and those may not reflect the negative class very well, especially when few supports are used. We suggest introducing a "null" class in prototypical models instead which does not require support examples. As batch normalization enforces zero centering in the latent space, we simply assume a zero centroid for this null class. We show that such models work better than the two-way implementation. Additionally, training time is reduced almost by a factor of two as only half of the support examples are required.

As another novel idea, we suggest modeling class prototypes not just by their mean (or centroid), but by their distribution. We assume normal distributions here, and model them via their mean and standard deviation. In the one-way case, we model the null class with a multivariate unit Gaussian (with mean 0 and standard deviation 1). This has the disadvantage of requiring a sufficient number of support examples to estimate these parameters. If those are available, the approach generally works, but usually does not surpass the standard prototypical model below a certain number of support examples (around 15).

We will further investigate whether these models can be useful when a very large number of support examples are available. There also is some evidence that such models train faster than standard prototypical models, which we will analyze. We suspect that one problem is the assumption of Gaussian-ness, which is not guaranteed in the latent space. It will be helpful to examine whether we can either enforce this, or otherwise find better ways to model the class distribution.

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

## A    APPENDIX

You may include other additional sections here.

