# OpenReview forum: "One-way prototypical networks"
_ICLR.cc/2020/Conference — Reject_

### Official Review · AnonReviewer3 · 2019-10-23
**Official Blind Review #3**

**Rating:** 8

**Review:**

This paper looks at the problem of few-shot classification in the regime when only a single class is present. The task at hand is as follows: given a number of support images of a previously unseen class (not present during training) and a single unlabeled image, we need to decide if this image belongs to the class or not. While previous approaches would explicitly construct negative examples to contrast the positive ones with during training, the authors bypass this by using batch norm in the last layer, which, on average, centers embedding feature vectors at 0, defining effectively the embedding for the negative class. In addition, the authors look at modelling the distribution of support image embeddings to improve the performance of their model.

Overview:
I like the general idea and find the paper easy to follow. Using the centering effect of batch norm to effectively bypass the need to define the universal negative class embedding is an interesting approach. I think it is especially compelling due to the special role of the \vec{0} point in the embedding space, where all embedding vectors of length 0, regardless of their direction, meet. I also appreciate that the authors used multiple training runs and their averages as their results, rather than the best result.

I have a few points that I believe should be clarified:

-- Point 1 --
Are you assuming that the trainable offset in the batch norm will be close to \vec{0}, or are you identically setting it to \vec{0}? Since your definition of the “garbage” class relies on this, it might be worth checking what the actual learned parameters in the batch norm end up being, and if they are small, possibly fixing them to 0?

-- Point 2 --
I didn’t get what distance metric in particular you were using to determine if the unlabeled example X is closer to the centroid of the class c or the origin 0. If I understood it correctly, you effectively look at length(c-X) > length(X-0) as your condition determining if X belongs to c. What metric did you use as your length measure?

-- Point 3 --
When using the Gaussian approximation estimating the distribution of the support images in the embedding space, how does it affect the condition  length(c-X) > length(X-0)? In “Gaussian Prototypical Networks for Few-Shot Learning on Omniglot” by Stanislav Fort you cite, they incorporate the covariance of the Gaussian S in the length function, effectively using length(c,X,S). Their Euclidean metric uses the covariance matrix is a metric tensor to weight different axes differently. How exactly do you use yours?

-- Point 4 --
When comparing to the previous algorithms in the single-class (one-way) regime, you use a random image from one (at random) of the C_train - 1 remaining training classes as your negative example / example of the other class. Is that a fair comparison? How many images of those classes do you use? If it were the case that for a problem with the support size of 1 (or a few), sampling from these C-1 other classes might provide very unrepresentative embedding vector for the negative class. Might it be fairer to look at the average embedding vector over more images from the negative classes?

-- Typos --
= 2 typos on page 4 at the bottom: “*Is* is then taken into account *explictly* in the distance metric.”
= 1 typo on page 5 at the bottom: “standard deviation between runs is generally very low (¡.01). Each train”

-- Conclusion --
Overall, I think the problem and solution proposed are interesting. The authors provide a large enough set of comparisons to existing algorithms, and explain their approach well.


**Experience Assessment:**

I have published one or two papers in this area.

**Review Assessment: Checking Correctness Of Derivations And Theory:**

N/A

**Review Assessment: Checking Correctness Of Experiments:**

I assessed the sensibility of the experiments.

**Review Assessment: Thoroughness In Paper Reading:**

I read the paper at least twice and used my best judgement in assessing the paper.

---

### Official Review · AnonReviewer2 · 2019-10-23
**Official Blind Review #2**

**Rating:** 3

**Review:**

Authors consider the 1-way few shot classification task. Argue that modeling it as a 2-way with a random negative sample is not efficient. Propose a novel technique applicable for prototypical networks. Their proposal is to use 0 as the prototype of null class. So the distance to the prototype is compared against the norm of the query embedding. They also propose modeling the distribution and not just the centroid using a multivariate gaussian. But in practice there is no benefit in doing so. Therefore, the main contribution is proposing to compare against norm of the embedding rather than a prototype for random negative samples. The benefit of this proposal decreases by more shots. Probably because the prototype of 20 random images is 0 anyway.

Fig. 4b is wrong. The 2-way results match the matching result in Fig. 3b rather than the prototypical result.

Considering Fig. 4 seems the most gain comes from the reordering rather than one-way proposal. So a natural question is what is the effect of reordering on the 2-way prototypical. It may boost the 2-way one too.

The contribution of this paper is intuitive and interesting. But given the experiments it seems insignificant. Specially with the missing reordering of the 2-way model results.

**Experience Assessment:**

I have read many papers in this area.

**Review Assessment: Checking Correctness Of Derivations And Theory:**

N/A

**Review Assessment: Checking Correctness Of Experiments:**

I carefully checked the experiments.

**Review Assessment: Thoroughness In Paper Reading:**

I read the paper thoroughly.

---

### Official Review · AnonReviewer1 · 2019-10-25
**Official Blind Review #1**

**Rating:** 3

**Review:**

This paper addresses a method of applying prototypical networks (which are popular for few-shot learning problems) to few-shot one-classification problems where only one group of examples are available without any counter-examples. The main idea of prototypical networks is to learn an embedding function such that in the embedding space a distance metric well reflects the class structure. When such models are applied to one-class problems, a basis for comparison is required.
The idea proposed in this paper is to introduce a null class which models the entire space of possible examples and queries are judged, compared to positive class as well as the null class. Another extension was made to model classes by their distribution instead of prototypes only, leading to one-way normal prototypical models.

---Strength---
- One-class extension of the prototypical network is a new idea, while prototypical networks are already popular for few-shot learning problems.
- The proposed model can solve few-shot anomaly detection problems.

---Weakness---
- Regarding the null class, it was claimed that the centroid of the whole latent space can be assumed to be close to the origin in the embedding space, since batch normalizations are included in the model.  The batch normalization learns the scale and shift parameters to minimize the error function, so the center does not have to be close to zero in all dimensions. More clear justification should be added to make the paper more sense.
- Regarding one-way normal prototypical models, calculating the distributions require sufficient number of examples. Thus, when applied to few-shot anomaly detection, the performance might be be satisfactory.


**Experience Assessment:**

I have published one or two papers in this area.

**Review Assessment: Checking Correctness Of Derivations And Theory:**

I carefully checked the derivations and theory.

**Review Assessment: Checking Correctness Of Experiments:**

I assessed the sensibility of the experiments.

**Review Assessment: Thoroughness In Paper Reading:**

I read the paper at least twice and used my best judgement in assessing the paper.

---

### Decision · Program_Chairs · 2019-12-19

**Decision:**

Reject

**Comment:**

This paper extends prototypical networks to few shot 1-way classification. The idea is to introduce a null class to compare against with a null prototype. The reviewers found the idea sound and interesting. However, the response was mixed because the reviewers were not convinced of the significance of the improvements. Furthermore, there were questions raised about the motivation that were not sufficiently addressed in the rebuttal. Batch normalization layers will not necessarily lead to zero mean if the trainable offset is not disabled. The authors did not clarify whether they disable this offset. I encourage the authors to resubmit after addressing the issues raised by the reviewers.